# Graph Convolutional Network with Sequential Attention For Goal-Oriented Dialogue Systems

## Abstract

Domain specific goal-oriented dialogue systems typically require modeling three types of inputs, *viz.*, (i) the knowledge-base associated with the domain, (ii) the history of the conversation, which is a sequence of utterances and (iii) the current utterance for which the response needs to be generated. While modeling these inputs, current state-of-the-art models such as Mem2Seq typically ignore the rich structure inherent in the knowledge graph and the sentences in the conversation context. Inspired by the recent success of structure-aware Graph Convolutional Networks (GCNs) for various NLP tasks such as machine translation, semantic role labeling and document dating, we propose a memory augmented GCN for goal-oriented dialogues. Our model exploits (i) the entity relation graph in a knowledge-base and (ii) the dependency graph associated with an utterance to compute richer representations for words and entities. Further, we take cognizance of the fact that in certain situations, such as, when the conversation is in a code-mixed language, dependency parsers may not be available. We show that in such situations we could use the global word co-occurrence graph and use it to enrich the representations of utterances. We experiment with the modified DSTC2 dataset and its recently released code-mixed versions in four languages and show that our method outperforms existing state-of-the-art methods, using a wide range of evaluation metrics.

## 1 Introduction

Goal-oriented dialogue systems which can assist humans in various day-to-day activities have widespread applications in several domains such as e-commerce, entertainment, healthcare, *etc*. For example, such systems can help humans in scheduling medical appointments, reserving restaurants, booking tickets, *etc.*. From a modeling perspective, one clear advantage of dealing with domain specific goal-oriented dialogues is that the vocabulary is typically limited, the utterances largely follow a fixed set of templates and there is an associated domain knowledge which can be exploited. More specifically, there is some structure associated with the utterances as well as the knowledge base.

More formally, the task here is to generate the next response given (i) the previous utterances in the conversation history (ii) the current user utterance (known as the query) and (iii) the entities and relationships in the associated knowledge base. Current state-of-the-art methods (Seo et al., 2017; Eric & Manning, 2017; Madotto et al., 2018) typically use variants of Recurrent Neural Network (Elman, 1990) to encode the history and current utterance and an external memory network to store the entities in the knowledge base. The encodings of the utterances and memory elements are then suitably combined using an attention network and fed to the decoder to generate the response, one word at a time. However, these methods do not exploit the structure in the knowledge base as defined by entity-entity relations and the structure in the utterances as defined by a dependency parse. Such structural information can be exploited to improve the performance of the system as demonstrated by recent works on syntax-aware neural machine translation (Eriguchi et al., 2016; Bastings et al., 2017; Chen et al., 2017), semantic role labeling (Marcheggiani & Titov, 2017) and document dating

(Vashishth et al., 2018) which use GCNs (Defferrard et al., 2016; Duvenaud et al., 2015; Kipf & Welling, 2017) to exploit sentence structure.

In this work, we propose to use such graph structures for goal-oriented dialogues. In particular, we compute the dependency parse tree for each utterance in the conversation and use a GCN to capture the interactions between words. This allows us to capture interactions between distant words in the sentence as long as they are connected by a dependency relation. We also use GCNs to encode the entities of the KB where the entities are treated as nodes and the relations as edges of the graph. Once we have a richer structure aware representation for the utterances and the entities, we use a sequential attention mechanism to compute an aggregated context representation from the GCN node vectors of the query, history and entities. Further, we note that in certain situations, such as, when the conversation is in a code-mixed language or a language for which parsers are not available then it may not be possible to construct a dependency parse for the utterances. To overcome this, we construct a co-occurrence matrix from the entire corpus and use this matrix to impose a graph structure on the utterances. More specifically, we add an edge between two words in a sentence if they co-occur frequently in the corpus. Our experiments suggest that this simple strategy acts as a reasonable substitute for dependency parse trees.

We perform experiments with the modified DSTC2 (Bordes et al., 2017) dataset which contains goal-oriented conversations for reserving restaurants. We also use its recently released code-mixed versions (Banerjee et al., 2018) which contain code-mixed conversations in four different languages, *viz.*, Hindi, Bengali, Gujarati and Tamil. We compare with recent state-of-the-art methods and show that on average the proposed model gives an improvement of 2.8 BLEU points and 2 ROUGE points.

Our contributions can be summarized as follows: (i) We use GCNs to incorporate structural information for encoding query, history and KB entities in goal-oriented dialogues (ii) We use a sequential attention mechanism to obtain query aware and history aware context representations (iii) We leverage co-occurrence frequencies and PPMI (positive-pointwise mutual information) values to construct contextual graphs for code-mixed utterances and (iv) We show that the proposed model obtains state-of-the-art results on the modified DSTC2 dataset and its recently released code-mixed versions.

## 2 RELATED WORK

In this section we review the previous work in goal-oriented dialogue systems and describe the introduction of GCNs in NLP.

**Goal-Oriented Dialogue System :** Initial goal-oriented dialogue systems (Young, 2000; Williams & Young, 2007) were based on dialogue state tracking (Williams et al., 2013; Henderson et al., 2014a;b) and included pipelined modules for natural language understanding, dialogue state tracking, policy management and natural language generation. Wen et al. (2017) used neural networks for these intermediate modules but still lacked absolute end-to-end trainability. Such pipelined modules were restricted by the fixed slot-structure assumptions on the dialogue state and required per-module based labelling. To mitigate this problem Bordes et al. (2017) released a version of goal-oriented dialogue dataset that focuses on the development of end-to-end neural models. Such models need to reason over the associated KB triples and generate responses directly from the utterances without any additional annotations. For example, Bordes et al. (2017) proposed a Memory Network (Sukhbaatar et al., 2015) based model to match the response candidates with the multi-hop attention weighted representation of the conversation history and the KB triples in memory. Liu & Perez (2017) further added highway (Srivastava et al., 2015) and residual connections (He et al., 2016) to the memory network in order to regulate the access to the memory blocks. Seo et al. (2017) developed a variant of RNN cell which computes a refined representation of the query over multiple iterations before querying the memory. However, all these approaches retrieve the response from a set of candidate responses and such a candidate set is not easy to obtain in any new domain of interest. To account for this, Eric & Manning (2017); Zhao et al. (2017) adapted RNN based encoder-decoder models to generate appropriate responses instead of retrieving them from a candidate set. Eric et al. (2017) introduced a key-value memory network based generative model which integrates the underlying KB with RNN based encode-attend-decode models. Madotto et al. (2018) used memory networks on top of the RNN decoder to tightly integrate KB entities with the decoder to generate more infor-

mative responses. However, as opposed to our work, all these works ignore the underlying structure of the entity-relation graph of the KB and the syntactic structure of the utterances.

**GCNs in NLP :** Recently, there has been an active interest in enriching existing encode-attend-decode models (Bahdanau et al., 2015) with structural information for various NLP tasks. Such structure is typically obtained from the constituency and/or dependency parse of sentences. The idea is to treat the output of a parser as a graph and use an appropriate network to capture the interactions between the nodes of this graph. For example, Eriguchi et al. (2016) and Chen et al. (2017) showed that incorporating such syntactical structures as Tree-LSTMs in the encoder can improve the performance of Neural Machine Translation (NMT). Peng et al. (2017) use Graph-LSTMs to perform cross sentence n-ary relation extraction and show that their formulation is applicable to any graph structure and Tree-LSTMs can be thought of as a special case of it. In parallel, Graph Convolutional Networks (GCNs) (Duvenaud et al., 2015; Defferrard et al., 2016; Kipf & Welling, 2017) and their variants (Li et al., 2015) have emerged as state-of-the-art methods for computing representations of entities in a knowledge graph. They provide a more flexible way of encoding such graph structures by capturing multi-hop relationships between nodes. This has led to their adoption for various NLP tasks such as neural machine translation (Marcheggiani et al., 2018; Bastings et al., 2017), semantic role labeling (Marcheggiani & Titov, 2017), document dating (Vashishth et al., 2018) and question answering (Johnson, 2017; Nicola De Cao, 2018).

To the best of our knowledge ours is the first work that uses GCNs to incorporate dependency structural information and the entity-entity graph structure in a single end-to-end neural model for goal-oriented dialogue. This is also the first work that incorporates contextual co-occurrence information for code-mixed utterances, for which no dependency structures are available.

## 3 BACKGROUND

In this section we describe Graph Convolutional Networks (GCN) (Kipf & Welling, 2017) for undirected graphs and then describe their syntactic versions which work with directed labeled edges of dependency parse trees.

### 3.1 GCN FOR UNDIRECTED GRAPHS

Graph convolutional networks operate on a graph structure and compute representations for the nodes of the graph by looking at the neighbourhood of the node. $k$ layers of GCNs can be stacked to account for neighbours which are $k$-hops away from the current node. Formally, let $\mathcal{G} = (\mathcal{V}, \mathcal{E})$ be an undirected graph where $\mathcal{V}$ is the set of nodes (let $|\mathcal{V}| = n$) and $\mathcal{E}$ is the set of edges. Let $\mathcal{X} \in \mathbb{R}^{n \times m}$ be the input feature matrix with $n$ nodes and each node $\mathbf{x}_u (u \in \mathcal{V})$ is represented by an $m$-dimensional feature vector. The output of a 1-layer GCN is the hidden representation matrix $\mathcal{H} \in \mathbb{R}^{n \times d}$ where each $d$-dimensional representation of a node captures the interactions with its 1-hop neighbour. Each row of this matrix can be computed as:

$$\mathbf{h}_v = ReLU\left( \sum_{u \in \mathcal{N}(v)} (W\mathbf{x}_u + \mathbf{b}) \right) \quad , \quad \forall v \in \mathcal{V} \tag{1}$$

Here $W \in \mathbb{R}^{d \times m}$ is the model parameter matrix, $b \in \mathbb{R}^d$ is the bias vector and $ReLU$ is the rectified linear unit activation function. $\mathcal{N}(v)$ is the set of neighbours of node $v$ and is assumed to also include the node $v$ so that the previous representation of the node $v$ is also considered while computing the new hidden representation. To capture interactions with nodes which are multiple hops away, multiple layers of GCNs can be stacked together. Specifically, the representation of node $v$ after $k^{th}$ GCN layer can be formulated as:

$$\mathbf{h}_v^{k+1} = ReLU\left( \sum_{u \in \mathcal{N}(v)} (W^k \mathbf{h}_u^k + \mathbf{b}^k) \right) \quad , \quad \forall v \in \mathcal{V} \tag{2}$$

where $\mathbf{h}_u^k$ is the representation of the $u^{th}$ node in the $(k-1)^{th}$ GCN layer and $\mathbf{h}_u^1 = \mathbf{x}_u$.

## 3.2 SYNTACTIC GCN

In a directed labeled graph $\mathcal{G} = (\mathcal{V}, \mathcal{E})$, each edge between nodes $u$ and $v$ is represented by a triple $(u, v, L(u, v))$ where $L(u, v)$ is the associated edge label. Marcheggiani & Titov (2017) modified GCNs to operate over directed labeled graphs, such as the dependency parse tree of a sentence. For such a tree, in order to allow information to flow from head to dependents and vice-versa, they added inverse dependency edges from dependents to heads such as $(v, u, L(u, v)')$ to $\mathcal{E}$ and made the model parameters and biases label specific. In their formulation,

$$\mathbf{h}_v^{k+1} = ReLU\Big( \sum_{u \in \mathcal{N}(v)} (W_{L(u,v)}^k \mathbf{h}_u^k + \mathbf{b}_{L(u,v)}^k) \Big) \quad , \quad \forall v \in \mathcal{V} \tag{3}$$

Notice that unlike equation 2, equation 3 has parameters $W_{L(u,v)}^k$ and $\mathbf{b}_{L(u,v)}^k$ which are label specific. Suppose there are $L$ different labels, then this formulation will require $L$ weights and biases per GCN layer resulting in a large number of parameters. To avoid this, the authors use only three sets of weights and biases per GCN layer (as opposed to $L$) depending on the direction in which the information flows. More specifically, $W_{L(u,v)}^k = V_{dir(u,v)}^k$, where $dir(u, v)$ indicates whether information flows from $u$ to $v$, $v$ to $u$ or $u = v$. In this work, we also make $b_{L(u,v)}^k = b_{dir(u,v)}^k$ instead of having a separate bias per label. The final GCN formulation can thus be described as:

$$\mathbf{h}_v^{k+1} = ReLU\Big( \sum_{u \in \mathcal{N}(v)} (W_{dir(u,v)}^k \mathbf{h}_u^k + \mathbf{b}_{dir(u,v)}^k) \Big) \quad , \quad \forall v \in \mathcal{V} \tag{4}$$

## 4 MODEL

We first formally define the task of end-to-end goal-oriented dialogue generation. Each dialogue of $t$ turns can be viewed as a succession of user utterances $(U)$ and system responses $(S)$ and can be represented as: $(U_1, S_1, U_2, S_2, ..U_t, S_t)$. Along with these utterances, each dialogue is also accompanied by $e$ KB triples which are relevant to that dialogue and can be represented as: $(k_1, k_2, k_3, ..k_e)$. Each triple is of the form: $(entity_1, relation, entity_2)$. These triples can be represented in the form of a graph $\mathcal{G}_k = (\mathcal{V}_k, \mathcal{E}_k)$ where $\mathcal{V}$ is the set of all entities and each edge in $\mathcal{E}$ is of the form: $(entity_1, entity_2, relation)$ where $relation$ signifies the edge label. At any dialogue turn $i$, given the (i) dialogue history $H = (U_1, S_1, U_2, ..S_{i-1})$, (ii) the current user utterance as the query $Q = U_i$ and (iii) the associated knowledge graph $\mathcal{G}_k$, the task is to generate the current response $S_i$ which leads to a completion of the goal. As mentioned earlier, we exploit the graph structure in KB and the syntactic structure in the utterances to generate appropriate responses. Towards this end we propose a model with the following components for encoding these three types of inputs.

## 4.1 QUERY ENCODER

The query $Q = U_i$ is the $i^{th}$ (current) utterance in the dialogue and contains $|Q|$ tokens. We denote the embedding of the $i^{th}$ token in the query as $\mathbf{q}_i$ We first compute the contextual representations of these tokens by passing them through a bidirectional RNN:

$$\mathbf{b}_t = BiRNN_Q(\mathbf{b}_{t-1}, \mathbf{q}_t) \tag{5}$$

Now, consider the dependency parse tree of the query sentence denoted by $\mathcal{G}_Q = (\mathcal{V}_Q, \mathcal{E}_Q)$. We use a query specific GCN to operate on $\mathcal{G}_Q$, which takes $\{\mathbf{b}_i\}_{i=1}^{|Q|}$ as the input to the $1^{st}$ GCN layer. The node representation in the $k^{th}$ hop of the query specific GCN is computed as:

$$\mathbf{c}_v^{k+1} = ReLU\Big( \sum_{u \in \mathcal{N}(v)} (W_{dir(u,v)}^k \mathbf{c}_u^k + \mathbf{g}_{dir(u,v)}^k) \Big) \quad , \quad \forall v \in \mathcal{V}_Q \tag{6}$$

where $W_{dir(u,v)}^k, \mathbf{g}_{dir(u,v)}^k$ are edge direction specific query-GCN weights and biases for the $k^{th}$ hop and $\mathbf{c}_u^1 = \mathbf{b}_u$.

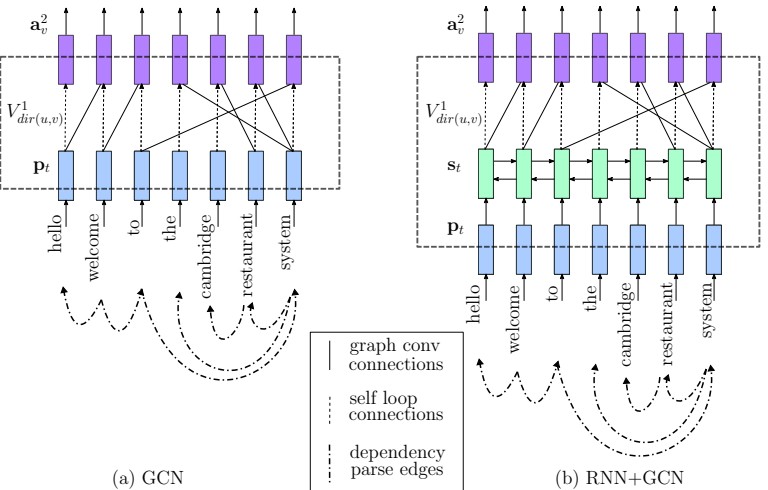

Figure 1: Illustration of the GCN and RNN+GCN modules which are used as encoders in our model. The notations are specific to the dialogue history encoder but both the encoders are same for the query. The GCN encoder is same for the KB except the graph structure.

## 4.2 DIALOGUE HISTORY ENCODER

The history $H$ of the dialogue contains $|H|$ tokens and we denote the embedding of the $i^{th}$ token in the history by $\mathbf{p}_i$ Once again, we first compute the hidden representations of these embeddings using a bidirectional RNN:

$$\mathbf{s}_t = BiRNN_H(\mathbf{s}_{t-1}, \mathbf{p}_t) \tag{7}$$

We now compute a dependency parse tree for each sentence in the history and collectively represent all the trees as a single graph $\mathcal{G}_H = (\mathcal{V}_H, \mathcal{E}_H)$. Note that this graph will only contain edges between words belonging to the same sentence and there will be no edges between words across sentences. We then use a history specific GCN to operate on $\mathcal{G}_H$ which takes $\mathbf{s}_t$ as the input to the $1^{st}$ layer. The node representation in the $k^{th}$ hop of the history specific GCN is computed as:

$$\mathbf{a}_v^{k+1} = ReLU\left( \sum_{u \in \mathcal{N}(v)} (V_{dir(u,v)}^k \mathbf{a}_u^k + \mathbf{o}_{dir(u,v)}^k) \right) \quad , \quad \forall v \in \mathcal{V}_H \tag{8}$$

where $V_{dir(u,v)}^k$ and $\mathbf{o}_{dir(u,v)}^k$ are edge direction specific history-GCN weights and biases in the $k^{th}$ hop and $\mathbf{a}_u^1 = \mathbf{s}_u$. Such an encoder with a single hop of GCN is illustrated in figure 1(b) and the encoder without the BiRNN is depicted in figure 1(a).

## 4.3 KB ENCODER

As mentioned earlier, $\mathcal{G}_K = (\mathcal{V}_K, \mathcal{E}_K)$ is the graph capturing the interactions between the entities in the knowledge graph associated with the dialogue. Let there be $m$ such entities and we denote the embeddings of the node corresponding to the $i^{th}$ entity as $\mathbf{e}_i$ We then operate a KB specific GCN on these entity representations to obtain refined representations which capture relations between entities. The node representation in the $k^{th}$ hop of the KB specific GCN is computed as:

$$\mathbf{r}_v^{k+1} = ReLU\left( \sum_{u \in \mathcal{N}(v)} (U_{dir(u,v)}^k \mathbf{r}_u^k + \mathbf{z}_{dir(u,v)}^k) \right) \quad , \quad \forall v \in \mathcal{V}_K \tag{9}$$

where $U_{dir(u,v)}^k$ and $\mathbf{z}_{dir(u,v)}^k$ are edge direction specific KB-GCN weights and biases in $k^{th}$ hop and $\mathbf{r}_u^1 = \mathbf{e}_u$. We also add inverse edges to $\mathcal{E}_K$ similar to the case of syntactic GCNs in order to allow information flow in both the directions for an entity pair in the knowledge graph.

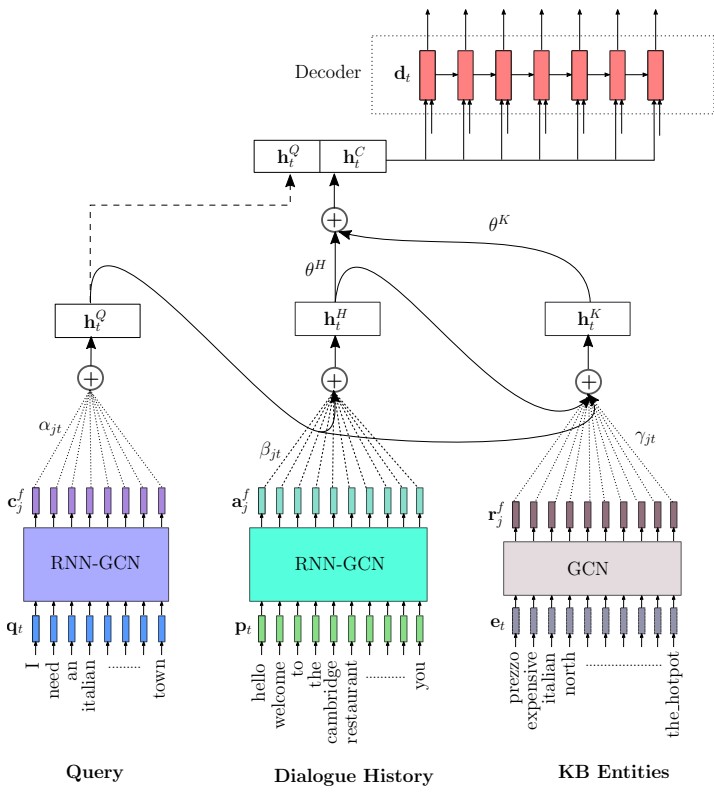

Figure 2: Illustration of sequential attention mechanism in RNN+GCN-SeA.

## 4.4 SEQUENTIAL ATTENTION

We use an RNN decoder to generate the tokens of the response and let the hidden states of the decoder be denoted as: $\{\mathbf{d}_i\}_{i=1}^{T}$ where $T$ is the total number of decoder timesteps. In order to obtain a single representation from the final layer ($k = f$) of the query-GCN node vectors, we use an attention mechanism as described below:

$$\mu_{jt} = \mathbf{v}_1 tanh(W_1 \mathbf{c}_j^f + W_2 \mathbf{d}_{t-1}) \tag{10}$$

$$\boldsymbol{\alpha}_t = \text{softmax}(\boldsymbol{\mu}_t) \tag{11}$$

$$\mathbf{h}_t^Q = \sum_{j'=1}^{|Q|} \alpha_{j't} \mathbf{c}_{j'}^f \tag{12}$$

Here $\mathbf{v}_1, W_1, W_2$ are parameters. Further, at each decoder timestep, we obtain a query aware representation from the final layer of the history-GCN by computing an attention score for each node/token in the history based on the query context vector $\mathbf{h}_t^Q$ as shown below:

$$\nu_{jt} = \mathbf{v}_2 tanh(W_3 \mathbf{a}_j^f + W_4 \mathbf{d}_{t-1} + W_5 \mathbf{h}_t^Q) \tag{13}$$

$$\boldsymbol{\beta}_t = \text{softmax}(\boldsymbol{\nu}_t) \tag{14}$$

$$\mathbf{h}_t^H = \sum_{j'=1}^{|H|} \beta_{j't} \mathbf{a}_{j'}^f \tag{15}$$

Here $\mathbf{v}_2, W_3, W_4$ and $W_5$ are parameters. Finally, we obtain a query and history aware representation of the KB by computing an attention score over all the nodes in the final layer of KB-GCN using $\mathbf{h}_t^Q$ and $\mathbf{h}_t^H$ as shown below:

$$\omega_{jt} = \mathbf{v}_3 tanh(W_6 \mathbf{r}_j^f + W_7 \mathbf{d}_{t-1} + W_8 \mathbf{h}_t^Q + W_9 \mathbf{h}_t^H) \tag{16}$$

$$\boldsymbol{\gamma}_t = \text{softmax}(\boldsymbol{\omega}_t) \tag{17}$$

$$\mathbf{h}_t^K = \sum_{j'=1}^{m} \gamma_{j't} \mathbf{r}_{j'}^f \tag{18}$$

Here $\mathbf{v}_3, W_6, W_7, W_8$ and $W_9$ are parameters. This sequential attention mechanism is illustrated in figure 2. For simplicity, we depict the GCN and RNN+GCN encoders as blocks. The internal structure of these blocks are shown in figure 1.

## 4.5 DECODER

The decoder takes two inputs, *viz.*, (i) the context which contains the history and the KB and (ii) the query which is the last/previous utterance in the dialogue. We use an aggregator which learns the overall attention to be given to the history and KB components. These attention scores: $\theta_t^H$ and $\theta_t^K$ are dependent on the respective context vectors and the previous decoder state $\mathbf{d}_{t-1}$. The final context vector is obtained as:

$$\mathbf{h}_t^C = \theta_t^H \mathbf{h}_t^H + \theta_t^K \mathbf{h}_t^K \qquad (19)$$

$$\mathbf{h}_t^{final} = [\mathbf{h}_t^C; \mathbf{h}_t^Q] \qquad (20)$$

where $[;]$ denotes the concatenation operator. At every timestep the decoder then computes a probability distribution over the vocabulary using the following equations:

$$\mathbf{d}_t = RNN(\mathbf{d}_{t-1}, [\mathbf{h}_t^{final}; \mathbf{w}_t]) \qquad (21)$$

$$P_{vocab} = \text{softmax}(V'\mathbf{d}_t + \mathbf{b}') \qquad (22)$$

where $\mathbf{w}_t$ is the decoder input at time step t, $V'$ and $\mathbf{b}'$ are parameters. $P_{vocab}$ gives us a probability distribution over the entire vocabulary and the loss for time step $t$ is $l_t = -\log P_{vocab}(w_t^*)$, where $w_t^*$ is the $t^{th}$ word in the ground truth response. The total loss is an average of the per-time step losses.

## 4.6 CONTEXTUAL GRAPH CREATION

For the dialogue history and query encoder, we used the dependency parse tree for capturing structural information in the encodings. However, if the conversations occur in a language for which no dependency parsers exist, for example: code-mixed languages like Hinglish (Hindi-English) (Banerjee et al., 2018) , then we need an alternate way of extracting a graph structure from the utterances. One simple solution which worked well in practice was to create a word co-occurrence matrix from the entire corpus where the context window is an entire sentence. Once we have such a co-occurrence matrix, for a given sentence we can connect an edge between two words if their co-occurrence frequency is above a threshold value. The co-occurrence matrix can either contain co-occurrence frequency counts or positive-pointwise mutual information (PPMI) values (Church & Hanks, 1990; Dagan et al., 1993; Niwa & Nitta, 1994).

## 5 EXPERIMENTAL SETUP

In this section we describe the datasets used in our experiments, the various hyperparameters that we considered and the models that we compared.

## 5.1 DATASETS

The original DSTC2 dataset (Henderson et al., 2014a) was based on the task of restaurant reservation and contains transcripts of real conversations between humans and bots. The utterances were labeled with the dialogue state annotations like the semantic intent representation, requested slots and the constraints on the slot values. We report our results on the modified DSTC2 dataset of Bordes et al. (2017) where such annotations are removed and only the raw utterance-response pairs are present with an associated set of KB triples for each dialogue. For our experiments with contextual graphs we reported our results on the code-mixed versions of modified DSTC2, which was recently released by Banerjee et al. (2018) [1]. This dataset has been collected by code-mixing the utterances of the English version of modified DSTC2 in four languages *viz.* Hindi (Hi-DSTC2), Bengali (Be-DSTC2), Gujarati (Gu-DSTC2) and Tamil (Ta-DSTC2), via crowdsourcing. Statistics about this dataset and example dialogues are shown in Appendix A.

---

[1] https://github.com/sumanbanerjee1/Code-Mixed-Dialog

| Model | per-resp. acc | BLEU | ROUGE | | | Entity F1 |
|---|---|---|---|---|---|---|
| | | | 1 | 2 | L | |
| Rule-Based (Bordes et al., 2017) | 33.3 | - | - | - | - | - |
| MEMNN (Bordes et al., 2017) | 41.1 | - | - | - | - | - |
| QRN (Seo et al., 2017) | 50.7 | - | - | - | - | - |
| GMEMNN (Liu & Perez, 2017) | 48.7 | - | - | - | - | - |
| Seq2Seq-Attn (Bahdanau et al., 2015) | 46.0 | 57.3 | 67.2 | 56.0 | 64.9 | 67.1 |
| Seq2Seq-Attn+Copy (Eric & Manning, 2017) | 47.3 | 55.4 | - | - | - | 71.6 |
| HRED (Serban et al., 2016) | 48.9 | 58.4 | 67.9 | 57.6 | 65.7 | 75.6 |
| Mem2Seq (Madotto et al., 2018) | 45.0 | 55.3 | - | - | - | 75.3 |
| GCN-SeA | 47.1 | 59.0 | 67.4 | 57.1 | 65.0 | 71.9 |
| RNN+CROSS-GCN-SeA | 51.2 | 60.9 | 69.4 | 59.9 | 67.2 | **78.1** |
| RNN+GCN-SeA | **51.4** | **61.2** | **69.6** | **60.2** | **67.4** | 77.9 |

Table 1: Comparison of GCN-SeA with other models on English version of modified DSTC2

| Dataset | Model | per-resp. acc | BLEU | ROUGE | | | Entity F1 |
|---|---|---|---|---|---|---|---|
| | | | | 1 | 2 | L | |
| Hi-DSTC2 | Seq2Seq-Bahdanau Attn | 48.0 | 55.1 | 62.9 | 52.5 | 61.0 | 74.3 |
| | HRED | 47.2 | 55.3 | 63.4 | 52.7 | 61.5 | 71.3 |
| | Mem2Seq | 43.1 | 50.2 | 55.5 | 48.1 | 54.0 | 73.8 |
| | GCN-SeA | 47.0 | 56.0 | 65.0 | 55.3 | 63.0 | 72.4 |
| | RNN+CROSS-GCN-SeA | 47.2 | 56.4 | 64.7 | 54.9 | 62.6 | 73.5 |
| | RNN+GCN-SeA | **49.2** | **57.1** | **66.4** | **56.8** | **64.4** | **75.9** |
| Be-DSTC2 | Seq2Seq-Bahdanau Attn | **50.4** | 55.6 | 67.4 | 57.6 | 65.1 | **76.2** |
| | HRED | 47.8 | 55.6 | 67.2 | 57.0 | 64.9 | 71.5 |
| | Mem2Seq | 41.9 | 52.1 | 58.9 | 50.8 | 57.0 | 73.2 |
| | GCN-SeA | 47.1 | 58.4 | 67.4 | 57.3 | 64.9 | 69.6 |
| | RNN+CROSS-GCN-SeA | **50.4** | 59.1 | 68.3 | 58.9 | 65.9 | 74.9 |
| | RNN+GCN-SeA | 50.3 | **59.2** | **69.0** | **59.4** | **66.6** | 75.1 |
| GU-DSTC2 | Seq2Seq-Bahdanau Attn | 47.7 | 54.5 | 64.8 | 54.9 | 62.6 | 71.3 |
| | HRED | 48.0 | 54.7 | 65.4 | 55.2 | 63.3 | 71.8 |
| | Mem2Seq | 43.1 | 48.9 | 55.7 | 48.6 | 54.2 | **75.5** |
| | GCN-SeA | 48.1 | 55.7 | 65.5 | 56.2 | 63.5 | 72.2 |
| | RNN+CROSS-GCN-SeA | **49.4** | **56.9** | **66.4** | **57.2** | **64.3** | 73.4 |
| | RNN+GCN-SeA | 48.9 | 56.7 | 66.1 | 56.9 | 64.1 | 73.0 |
| Ta-DSTC2 | Seq2Seq-Bahdanau Attn | 49.3 | 62.9 | 67.8 | 56.3 | 65.6 | 77.7 |
| | HRED | 47.8 | 61.5 | 66.9 | 55.2 | 64.8 | 74.4 |
| | Mem2Seq | 44.2 | 58.9 | 58.6 | 50.8 | 57.0 | 74.9 |
| | GCN-SeA | 46.4 | 62.8 | 68.5 | 57.5 | 66.1 | 71.9 |
| | RNN+CROSS-GCN-SeA | **50.8** | 64.5 | 69.8 | 59.6 | 67.5 | **78.8** |
| | RNN+GCN-SeA | 50.7 | **64.9** | **70.2** | **59.9** | **67.9** | 77.9 |

Table 2: Comparison of RNN+GCN-SeA, GCN-SeA with other models on all code-mixed datasets

## 5.2 HYPERPARAMETERS

We used the same train, test and validation splits as provided in the original versions of the datasets. We minimized the cross entropy loss using the Adam optimizer (Kingma & Ba, 2015) and tuned the initial learning rates in the range of 0.0006 to 0.001. For regularization we used an L2 norm of 0.001 in addition to a dropout (Srivastava et al., 2014) of 0.1. We used randomly initialized word embeddings of size 300. The RNN and GCN hidden dimensions were also chosen to be 300. We use GRU (Cho et al., 2014) cells for the RNNs. All parameters were initialized from a truncated normal distribution with a standard deviation of 0.1.

## 5.3 MODELS COMPARED

We compare the performance of the following models.

**(i) RNN+GCN-SeA vs GCN-SeA :** We use RNN+GCN-SeA to refer to the model described in section 4. Instead of using the hidden representations obtained from the bidirectional RNNs, we also experiment by providing the token embeddings directly to the GCNs i.e. $\mathbf{c}_u^1 = \mathbf{q}_u$ in equation 6 and $\mathbf{a}_u^1 = \mathbf{p}_u$ in equation 8. We refer to this model as GCN-SeA.

**(ii) Cross edges between the GCNs:** In addition to the dependency and contextual edges, we add edges between words in the dialogue history/query and KB entities if a history/query word exactly matches the KB entity. Such edges create a single connected graph which is encoded using a single GCN encoder and then separated into different contexts to perform the sequential attention. This model is referred to as RNN+CROSS-GCN-SeA.

**(iii) Frequency vs PPMI Contextual Graph :** We experiment with the raw frequency co-occurrence graph structure and the PPMI graph structure for the code-mixed datasets, as explained in section 4.6. We refer to these models as GCN-SeA+Freq and GCN-SeA+PPMI. In both these models, the GCN takes inputs from a bidirectional RNN.

**(iv) GCN-SeA+Random vs GCN-SeA+Structure :** We experiment with the model where the graph is constructed by randomly connecting edges between two words in a context. We refer to this model as GCN-SeA+Random. We refer to the model which either uses dependency or contextual graph instead of random graphs as GCN-SeA+Structure.

## 6 RESULTS AND DISCUSSIONS

In this section we discuss the results of our experiments as summarized in tables 1,2, and 3. We use BLEU (Papineni et al., 2002) and ROUGE (Lin, 2004) metrics to evaluate the generation quality of responses. We also report the per-response accuracy which computes the percentage of responses in which the generated response exactly matches the ground truth response. In order to evaluate the model's capability of correctly injecting entities in the generated response, we report the entity F1 measure as defined in Eric & Manning (2017).

**Results on En-DSTC2 :** We compare our model with the previous works on the English version of modified DSTC2 in table 1. For most of the retrieval based models, the BLEU or ROUGE scores are not available as they select a candidate from a list of candidates as opposed to generating it. Our model outperforms all of the retrieval and generation based models. We obtain a gain of 0.7 in the per-response accuracy compared to the previous retrieval based state-of-the-art model of Seo et al. (2017), which is a very strong baseline for our generation based model. We call this a strong baseline because the candidate selection task of this model is easier than the response generation task of our model. We also obtain a gain of 2.8 BLEU points, 2 ROUGE points and 2.5 entity F1 points compared to current state-of-the-art generation based models.

**Results on code-mixed datasets and effect of using RNNs:** The results of our experiments on the code-mixed datasets are reported in table 2. Our model outperforms the baseline models on all the code-mixed languages. One common observation from the results over all the languages (including En-DSTC2) is that RNN+GCN-SeA performs better than GCN-SeA. Similar observations were made by Marcheggiani & Titov (2017) for the task of semantic role labeling.

**Effect of using Hops:** As we increased the number of hops of GCNs, we observed a decrease in the performance. One reason for such a drop in performance could be that the average utterance length is very small (7.76 words). Thus, there isn't much scope for capturing distant neighbourhood information and more hops can add noisy information. Please refer to Appendix B for detailed results about the effect of varying the number of hops.

**Frequency vs PPMI graphs:** We observed that PPMI based contextual graphs were slightly better than frequency based contextual graphs (See Appendix C). In particular, when using PPMI as opposed to frequency based contextual graph, we observed a gain of 0.95 in per-response accuracy, 0.45 in BLEU, 0.64 in ROUGE and 1.22 in entity F1 score when averaged across all the code-mixed languages.

**Effect of using Random Graphs:** GCN-SeA-Random and GCN-SeA-Structure take the token embeddings directly instead of passing them though an RNN. This ensures that the difference in performance of the two models are not influenced by the RNN encodings. The results are shown in table 3 and we observe a drop in performance for GCN-Random across all the languages. This

| Dataset | Model | per-resp. acc | BLEU | ROUGE | | | Entity F1 |
|---------|-------|------|------|---|---|---|-----------|
| | | | | 1 | 2 | L | |
| En-DSTC2 | GCN-SeA+Random | 45.9 | 57.8 | 67.1 | 56.5 | 64.8 | **72.2** |
| | GCN-SeA+Structure | **47.1** | **59.0** | **67.4** | **57.1** | **65.0** | 71.9 |
| Hi-DSTC2 | GCN-SeA+Random | 44.4 | 54.9 | 63.1 | 52.9 | 60.9 | 67.2 |
| | GCN-SeA+Structure | **47.0** | **56.0** | **65.0** | **55.3** | **63.0** | **72.4** |
| Be-DSTC2 | GCN-SeA+Random | 44.9 | 56.5 | 65.4 | 54.8 | 62.7 | 65.6 |
| | GCN-SeA+Structure | **47.1** | **58.4** | **67.4** | **57.3** | **64.9** | **69.6** |
| Gu-DSTC2 | GCN-SeA+Random | 45.0 | 54.0 | 64.1 | 54.0 | 61.9 | 69.1 |
| | GCN-SeA+Structure | **48.1** | **55.7** | **65.5** | **56.2** | **63.5** | **72.2** |
| Ta-DSTC2 | GCN-SeA+Random | 44.8 | 61.4 | 66.9 | 55.6 | 64.3 | 70.5 |
| | GCN-SeA+Structure | **46.4** | **62.8** | **68.5** | **57.5** | **66.1** | **71.9** |

Table 3: GCN-SeA with random graphs and frequency co-occurrence graphs on all DSTC2 datasets.

shows that any random graph does not contribute to the performance gain of GCN-SeA and the dependency and contextual structures do play an important role.

**Ablations :** We experiment with replacing the sequential attention by the Bahdanau attention (Bahdanau et al., 2015). We also experiment with various combinations of RNNs and GCNs as encoders. The results are shown in table 8 (Appendix D). We observed that GCNs do not outperform RNNs independently. In general, RNN-Bahdanau attention performs better than GCN-Bahdanau attention. The sequential attention mechanism outperforms Bahdanau attention as observed from the following comparisons (i) GCN-Bahdanau attention vs GCN-SeA, (ii) RNN-Bahdanau attention vs RNN-SeA (in BLEU and ROUGE) and (iii) RNN+GCN-Bahdanau attention vs RNN+GCN-SeA. Overall, the best results are always obtained by our final model which combines RNN, GCN and sequential attention.

## 7 CONCLUSION

We showed that structure aware representations are useful in goal-oriented dialogue and we obtain state-of-the art performance on the modified DSTC2 dataset and its recently released code-mixed versions. We used GCNs to infuse structural information of dependency graphs and contextual graphs to enrich the representations of the dialogue context and KB. We also proposed a sequential attention mechanism for combining the representations of (i) query (current utterance), (ii) conversation history and (ii) the KB. Finally, we empirically showed that when dependency parsers are not available for certain languages such as code-mixed languages then we can use word co-occurrence frequencies and PPMI values to extract a contextual graph and use such a graph with GCNs for improved performance.

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

## A. CODE-MIXING STATISTICS AND DIALOGUE EXAMPLE

| | Hindi | Bengali | Gujarati | Tamil |
|---|---|---|---|---|
| Vocabulary Size | 1676 | 1372 | 1858 | 2185 |
| Code-Mixed English Vocabulary | 386 | 360 | 387 | 424 |
| Native Language Vocabulary | 739 | 477 | 912 | 1214 |
| Others Vocabulary | 551 | 535 | 559 | 547 |
| Unique Utterances | 6549 | 6274 | 6417 | 6666 |
| Utterances with code-mixed words | 5750 | 5703 | 5643 | 5632 |
| Pure Native Language utterances | 348 | 210 | 340 | 420 |
| Pure English utterances | 451 | 361 | 434 | 614 |
| Average length of utterances | 8.16 | 7.74 | 8.04 | 6.78 |
| Average # of code-mixed utterances per dialog | 12.11 | 14.28 | 11.80 | 12.96 |

Table 4: Statistics about code-mixing in the dataset (Banerjee et al., 2018).

| | English | Hindi-English codemixed |
|---|---|---|
| user | <SILENCE> | <SILENCE> |
| bot | Hello, welcome to the Cambridge restaurant system. You can ask for restaurants by area, price range or food type. How may I help you ? | *Hello , Cambridge restaurant system* mein aapka swagat hai. Aap pa sakte hai *restaurant by area, price range* ya *food type*. Aap ko kaise *help* kar sakta hu main ? |
| user | im looking for a moderately priced restaurant. | mein *moderately priced restaurant* dhund raha hoon. |
| bot | What kind of food would you like ? | Aap kaunse *type* ka *food* lena chahege? |
| user | it doesnt matter. | *matter* nahi karta. |
| bot | What part of town do you have in mind ? | aapke *mind* me *town* ka kaun sa *part* hai ? |
| user | the south part of town. | *south part of town*. |
| bot | api_call R_cuisine south moderate | api_call R_cuisine south moderate |
| | **KB Triples:** pizza_hut_cherry_hinton R_post_code pizza_hut_cherry_hinton_post_code pizza_hut_cherry_hinton R_cuisine italian pizza_hut_cherry_hinton R_location south pizza_hut_cherry_hinton R_phone pizza_hut_cherry_hinton_phone pizza_hut_cherry_hinton R_address pizza_hut_cherry_hinton_address pizza_hut_cherry_hinton R_price moderate pizza_hut_cherry_hinton R_rating 3 restaurant_alimentum R_post_code restaurant_alimentum_post_code restaurant_alimentum R_cuisine european restaurant_alimentum R_location south restaurant_alimentum R_phone restaurant_alimentum_phone restaurant_alimentum R_address restaurant_alimentum_address restaurant_alimentum R_price moderate restaurant_alimentum R_rating 10 | |
| user | <SILENCE> | <SILENCE> |
| bot | restaurant_alimentum is a nice restaurant in the south of town serving modern european food. | restaurant_alimentum ek bohot hi accha *restaurant* hai *south town* me jo *modern european food serve* karta hai |
| user | can i get the address ? | kya mujhe *address* mil sakta hai |
| bot | Sure, restaurant_alimentum is on restaurant_alimentum_address. | Bilkul, restaurant_alimentum restaurant_alimentum_address par hai |
| user | and whats the phone number for that ? | aur iska *phone number* kya hai ? |
| bot | The phone number of restaurant_alimentum is restaurant_alimentum_phone. | restaurant_alimentum ka *phone number* hai restaurant_alimentum_phone |
| user | okay, thank you, good bye. | theek, *thank you, good bye*. |
| bot | you are welcome. | *welcome*, alvida. |

Table 5: Example dialogues from the DSTC2 dataset in English and in Hindi-English code-mixed languages. The English words which are mixed in Hindi are *italicized*.

## B. Effect of using Hops:

| Dataset | Model | per-resp. acc | BLEU | ROUGE | | | Entity F1 |
|---|---|---|---|---|---|---|---|
| | | | | 1 | 2 | L | |
| En-DSTC2 | GCN-SeA+K=1 | 47.1 | 59.0 | 67.4 | 57.1 | 65.0 | 71.9 |
| | GCN-SeA+K=2 | **48.4** | **59.7** | **68.5** | **58.4** | **66.2** | **72.8** |
| | GCN-SeA+K=3 | 46.4 | 59.4 | 67.3 | 56.9 | 64.8 | 68.8 |
| Hi-DSTC2 | GCN-SeA+K=1 | **47.0** | **56.0** | **65.0** | **55.3** | **63.0** | **72.4** |
| | GCN-SeA+K=2 | 40.4 | 53.2 | 61.8 | 50.5 | 59.7 | 60.2 |
| | GCN-SeA+K=3 | 19.0 | 29.7 | 42.2 | 28.9 | 38.5 | 00.5 |
| Be-DSTC2 | GCN-SeA+K=1 | **47.1** | **58.4** | **67.4** | **57.3** | **64.9** | **69.6** |
| | GCN-SeA+K=2 | 41.9 | 55.2 | 64.5 | 53.5 | 61.9 | 61.4 |
| | GCN-SeA+K=3 | 07.0 | 25.6 | 34.3 | 16.8 | 25.0 | 02.4 |
| GU-DSTC2 | GCN-SeA+K=1 | **48.1** | **55.7** | **65.5** | **56.2** | **63.5** | **72.2** |
| | GCN-SeA+K=2 | 43.3 | 53.5 | 63.7 | 53.4 | 61.5 | 64.2 |
| | GCN-SeA+K=3 | 20.8 | 36.5 | 47.3 | 34.1 | 45.1 | 17.3 |
| Ta-DSTC2 | GCN-SeA+K=1 | **46.4** | **62.8** | **68.5** | **57.5** | **66.1** | **71.9** |
| | GCN-SeA+K=2 | 44.4 | 61.5 | 67.2 | 55.8 | 64.7 | 68.8 |
| | GCN-SeA+K=3 | 36.4 | 56.1 | 62.2 | 49.9 | 59.9 | 56.0 |

Table 6: GCN-SeA with multiple hops on all DSTC2 datasets

## C. Frequency vs PPMI co-occurrence

| Dataset | Model | per-resp. acc | BLEU | ROUGE | | | Entity F1 |
|---|---|---|---|---|---|---|---|
| | | | | 1 | 2 | L | |
| En-DSTC2 | GCN-SeA+Freq | 50.4 | **61.1** | 69.3 | 59.6 | 67.0 | 76.0 |
| | GCN-SeA+PPMI | **50.5** | 60.7 | 69.3 | **59.7** | 67.0 | **77.4** |
| Hi-DSTC2 | GCN-SeA+Freq | 48.7 | 56.9 | 65.5 | 56.1 | 63.5 | 74.5 |
| | GCN-SeA+PPMI | **49.2** | **57.1** | **66.4** | **56.8** | **64.4** | **75.9** |
| Be-DSTC2 | GCN-SeA+Freq | 49.0 | 59.0 | 68.2 | 58.5 | 65.7 | 72.7 |
| | GCN-SeA+PPMI | **50.3** | **59.2** | **69.0** | **59.4** | **66.6** | **75.1** |
| Gu-DSTC2 | GCN-SeA+Freq | 48.4 | 56.1 | **66.2** | 56.7 | 64.0 | **73.3** |
| | GCN-SeA+PPMI | **48.9** | **56.7** | 66.1 | **56.9** | **64.1** | 73.0 |
| Ta-DSTC2 | GCN-SeA+Freq | 49.2 | 64.1 | 69.5 | 59.0 | 67.1 | 76.7 |
| | GCN-SeA+PPMI | **50.7** | **64.9** | **70.2** | **59.9** | **67.9** | **77.9** |

Table 7: RNN+GCN-SeA with different contextual graphs on all DSTC2 datasets

## D. ABLATION RESULTS

| Dataset | Model | per-resp. acc | BLEU | ROUGE | | | Entity F1 |
|---|---|---|---|---|---|---|---|
| | | | | 1 | 2 | L | |
| Hi-DSTC2 | Seq2seq-Bahdanau Attn | 48.0 | 55.1 | 62.9 | 52.5 | 61.0 | 74.3 |
| | GCN-Bahdanau Attn | 38.5 | 50.4 | 58.9 | 47.7 | 56.7 | 59.1 |
| | RNN+GCN-Bahdanau Attn | 47.1 | 56.0 | 65.1 | 55.2 | 62.9 | 72.2 |
| | RNN-SeA | 45.8 | 55.9 | 65.1 | 55.5 | 63.1 | 71.8 |
| | RNN+GCN-SeA | **49.2** | **57.1** | **66.4** | **56.8** | **64.4** | **75.9** |
| Be-DSTC2 | Seq2seq-Bahdanau Attn | **50.4** | 55.6 | 67.4 | 57.6 | 65.1 | **76.2** |
| | GCN-Bahdanau Attn | 42.1 | 55.1 | 63.7 | 52.8 | 61.1 | 64.3 |
| | RNN+GCN-Bahdanau Attn | 47.0 | 57.7 | 67.0 | 57.4 | 64.6 | 70.9 |
| | RNN-SeA | 46.8 | 58.5 | 67.6 | 58.1 | 65.1 | 71.9 |
| | RNN+GCN-SeA | 50.3 | **59.2** | **69.0** | **59.4** | **66.6** | 75.1 |
| Gu-DSTC2 | Seq2seq-Bahdanau Attn | 47.7 | 54.5 | 64.8 | 54.9 | 62.6 | 71.3 |
| | GCN-Bahdanau Attn | 38.8 | 49.5 | 59.2 | 48.3 | 56.8 | 58.0 |
| | RNN+GCN-Bahdanau Attn | 46.5 | 55.5 | 65.6 | 55.9 | 63.4 | 70.6 |
| | RNN-SeA | 45.4 | 56.0 | 66.0 | 56.6 | 63.9 | 69.8 |
| | RNN+GCN-SeA | **48.9** | **56.7** | **66.1** | **56.9** | **64.1** | **73.0** |
| Ta-DSTC2 | Seq2seq-Bahdanau Attn | 49.3 | 62.9 | 67.8 | 56.3 | 65.6 | 77.7 |
| | GCN-Bahdanau Attn | 42.0 | 59.3 | 64.8 | 52.8 | 62.1 | 69.7 |
| | RNN+GCN-Bahdanau Attn | 46.3 | 63.2 | 68.0 | 57.2 | 65.6 | 72.1 |
| | RNN-SeA | 46.8 | 64.0 | 69.3 | 59.0 | 67.1 | 74.2 |
| | RNN+GCN-SeA | **50.7** | **64.9** | **70.2** | **59.9** | **67.9** | **77.9** |
| En-DSTC2 | Seq2seq-Bahdanau Attn | 46.0 | 57.3 | 67.2 | 56.0 | 64.9 | 67.1 |
| | GCN-Bahdanau Attn | 45.7 | 58.1 | 66.5 | 55.9 | 64.1 | 70.1 |
| | RNN+GCN-Bahdanau Attn | 47.4 | 59.5 | 67.9 | 57.7 | 65.6 | 72.9 |
| | RNN-SeA | 47.0 | 60.2 | 68.5 | 58.9 | 66.2 | 72.7 |
| | RNN+GCN-SeA | **51.4** | **61.2** | **69.6** | **60.2** | **67.4** | **77.9** |

Table 8: Ablation results of various models on all versions of DSTC2.

