# OpenReview forum: "Graph Convolutional Network with Sequential Attention For Goal-Oriented Dialogue Systems"
_ICLR.cc/2019/Conference_

### Official Review · AnonReviewer3 · 2018-10-22
**A dialogue system that is novel in using graph convolutional networks as part of an encoder-decoder dialogue system with attention**

**Rating:** 7
**Confidence:** 2

**Review:**

This is a well-written paper (especially the introduction) with fairly extensive experimentation section. It'a very possitive for me that you resort to more than one set of figures of merit.

My concerns are:

You mention that GCNs have been used for question-anwering already. It would be infomative to furhter describe this work and clearly state how you handle things differenclty, since a Q&A system is quite close to a dialogue one.

There are some parts that could be made more clear. For example, when you mention that you collectively represent all trees as a single graph. How do you do that?

The model has a great number of parameters. It is not clear to me how you concluded to the specific parameter values.

It would be nice to add the complexity of the model and also be more specific about how you choose the parameter values.

My proposals are:

I think that the paper would greatly benefit if you additionally to the equations you also presented the model in a graphical way as well. Additionally, although the paper is very well mathematically defined, is not so easy to follow from a practical perspective. For example, regarding section 5.3 I would prefer to see the 3 models you present in a graphical way as well.

Maybe add the links to the datasets you are using? On a related subject, would your models be transferable accross datasets?

Minor issues
PPMI abreviation is first used and then defined.
There are also some typos, like conisdered (that I suppose was meant to be considered, for example)

---

> ### Author Response · Authors · 2018-11-24
> **Authors’ response to Reviewer 3**
>
> We would like to thank you for your comments and valuable suggestions on improving the clarity of the paper. Below, we provide updates on some of the improvements that we have been able to do:
>
> 1) Difference from QA:  To the best of our knowledge, the only work on using GCNs for QA (De Cao et. al., 2018) uses Entity graphs as opposed to dependency graphs used in the work. An entity graph essentially draws an edge between same entities which appear in different sentences whereas a dependency graph contains semantic edges. Hence, the graph that we are operating on is very different from the entity graph used in the above paper. Further, in the case of QA, there is a query and a document whereas in our case there is a context/history in addition to the query and KB. This adds some more complexity to our model. For example, our sequential attention mechanism also considers the history while paying attention to the KB. Further, it also computes a query aware representation for the history. Finally, while producing the output, the decoder also pays attention to the history. These differences are not groundbreaking but we just mention them here to make the distinction between the two tasks clear and to highlight the additional components in our model.
>
> 2) How do we collectively represent all trees as a graph? This was in the context of computing a representation for the dialogue history. The history contains multiple sentences. We first create a dependency tree for each sentence. The final graph for the history is a simple collection of these individual (disconnected) trees. Just to be clear, currently we do not have any edges between words in two different sentences (hence all the individual sentence trees are disconnected from each other).
>
> 3) Regarding complexity and choosing parameter values: Our final model (RNN+GCN-SeA) has ~4M parameters as compared to the vanilla RNN+attention model which has ~2M parameters. These parameters were learned using ADAM, with a batch size of 32 and initial learning rate of 0.0006. We found that the model trains in ~30 epochs. In addition, we would like to clarify that the hyperparameters of the model were chosen using a validation set.
>
> 4) Clarity: Indeed, in hindsight and based on similar comments by Reviewer 1, we agree that we could have made things more clear by adding a diagram. We have included 2 diagrams in the updated version of the paper and we hope it clarifies things. Regarding the three models in Section 5.3, they would only differ in the type of parse tree edges (last edge type in the legend) shown in Figure 1. Please give us your feedback on the diagrams and if it can be improved to make things more clear.
>
> 5) Link to Dataset: We used the dataset released by Banerjee et. al., 2018 which is available at the following URL: https://github.com/sumanbanerjee1/Code-Mixed-Dialog . We plan to include results on the In-Car dataset also. We are hopeful that we will have these results ready in the final version of the paper.
>
> 6) Thanks for pointing out the typos. We have fixed them in the updated version of the paper.

---

> > ### Comment · AnonReviewer3 · 2018-11-26
> > **More clear paper**
> >
> > Many thanks for adding the figures - they have improved my understading of the paper and I think they make it more easy to understand.

---

### Official Review · AnonReviewer2 · 2018-11-02
**Good performance but less clear novelty**

**Rating:** 6
**Confidence:** 4

**Review:**

The paper proposes a Graph Convolutional Network-based encoder-decoder model with sequential attention for goal-oriented dialogue systems, with the purpose of exploiting the graph structures in KB and sentences in conversation. The model consists of three encoders for a query, dialogue history, and KB, respectively, and a decoder with a sequential attention mechanism. The proposed model attains state-of-the-art performance on the modified DSTC2 dataset of (Bordes et al., 2017). For the experiments with graphs constructed from word co-occurrence matrix, code-mixed versions of modified DSTC2 released by (Banerjee et al., 2018) are used.

Pros and Cons
(+) SOTA performance on the DSTC2 dataset.
(+) Without dependency parser when it is not possible
(-) Limited novelty
(-) Limited convincing the advantage of GCN itself

Detailed comments
The paper incorporates the graph structures in sentences and KB to make richer representations of conversation and achieves a state-of-the-art performance on the DSTC2 dataset. The paper is clearly written, and the results seem promising. However, as the paper combines existing mechanisms to design a model for dialog, the novelty seems to be relatively weak.
In particular, I felt that some experimental results are required to verify some of the arguments put forward by the authors. We listed two issues as below.

1. Effects of GCN
The authors show that RNN-GCN-SeA can make state-of-the-art performance, but not how much GCN makes effects on improving the performance on the dialog task.
I think the authors need to compare the results of RNN-GCN-SeA with a model without GCN (i.e. RNN-SeA) in order to show that exploiting the structural information of dependency and contextual graphs do play an important role.
The random graph experiments (Table 3) show the effect of good structure in GCN, but I felt that it is not enough to demonstrate an improvement by GCNs.

2. Comparative Experiments
I think that some experiments, which is reported in the previous papers (including Mem2Seq), would make the author’s experimental argument strong.
- Entity F1 score for the modified DSTC2 dataset
- Results on bAbI dialog dataset (task1~5 and its OOV variants) and In-Car Assistant dataset

Minor issues
1. Authors described that Mem2Seq is one of the state-of-the-art models in this field, including in the abstract. However, Mem2Seq does not outperform seq2seq model in all experiments. From what point of view is this model state-of-the-art?
2. Recent studies have focused on copy mechanism in task-oriented dialog systems. Could you explain how the copy mechanism could be incorporated into the proposed model? I am also interested in the comparative results between seq2seq + attn + copy (per-resp-acc of 47.3) and its entity F1 measure (Eric and Manning, 2017; Madotto et al. 2018).

---

> ### Author Response · Authors · 2018-11-24
> **Authors’ response to Reviewer 2**
>
> We would like to thank you for suggesting additional experiments for improving the paper. We have been able to run these experiments and would like to update you about the results:
>
> 1) Effects of GCN: We have now added detailed ablation studies (please see Table 8 in Appendix D) including comparisons with basic RNN based models and basic attention models. In particular, we have now compared RNN+GCN-SeA with RNN-SeA. The results indeed suggest that adding GCNs on top of RNNs helps. Our analysis also shows that our sequential attention outperforms the basic (Bahdanau) attention. Please see “Ablations” part of Section 6 and Table 8 in Appendix D. Also, the code for our model and these ablation studies will be made publicly available.
>
> 2) Comparative experiments: We have reported Entity F1 scores for all our experiments and again find that w.r.t this metric our model mostly outperforms existing approaches (including some new baselines that we have added for the ablation study). We were not very keen on the bAbI dataset since existing research ( Hybrid Code Networks, Williams et. al., 2017 ) shows that it is possible to achieve 100% performance on this dataset using simple models (not surprising given that this is a synthetic dataset). Hence, there is not much scope for introducing more complex models such as the one proposed in this paper. We plan to include results on the In-Car dataset and we are hopeful that we will have these results ready in the final version of the paper.
>
> 3) Yes, indeed, Mem2Seq does not outperform seq2seq in all experiments. In that sense, you are correct in saying that it is not a SOTA model. By SOTA, we incorrectly meant that it is the most recent model published on this dataset.
>
> 4) Copy mechanism: In addition to the Mem2Seq model of (Madotto et. al., 2018), we have now added the comparison with the model of (Eric and Manning, 2017) which uses copy mechanism. Our model outperforms both these models. In principle, we should be able to augment our model with a copy mechanism but this may be a non-trivial extension of our model. This is definitely worth trying but we are not sure if we will be able to add this to the current version of the paper. We apologize for this (we don't want to commit to something that we may not be able to deliver).

---

### Official Review · AnonReviewer1 · 2018-11-05
**Interesting topic, requires a somewhat better analysis**

**Rating:** 5
**Confidence:** 3

**Review:**

The current paper proposes using Graph Convolutional Networks (GCN) to explicitly represent and use relational data in dialog modeling, as well an attention mechanism for combining information from multiple sources (dialog history, knowledge base, current utterance). The work assumes that the knowledge base associated with the dialog task has en entity-to-entity-relationship format and can be naturally expressed as a graph. The dependency tree of dialog utterances can also be expressed as a graph, and the dialog history as a set of graphs. To utilize this structure, the proposed method uses GCNs whose lowest layer embeddings are initialized with the entity embeddings or via outputs of standard RNN-like models. The main claim is that the proposed model outperforms the current state-of-the-art on a goal-oriented dialog task.

The idea of explicitly modeling the relational structure via GCNs is interesting. However, the use of GCNs independently per sentence and per knowledge-base is a bit disappointing, since it does not couple these sources of information in a structured way. Instead, from my current understanding, the approach merely obtains better representations for each of these sources of information, in the same way it is done in the related language tasks. For instance, have you considered passing information across the trees in the history as well? Or aligning the parsed query elements with the KB elements?

The results are very good. That said, a source of concern is that the model is only evaluated as a whole, without showing which modification brought the improvements. The comparison between using/not using RNNs to initiate the first GCN layer is promising, but why not compare to using only RNN also? Why not compare the various encoders within an established framework (e.g. without the newly introduced attention mechanism)? Finally, the attention mechanism, stated as a contribution, is not motivated well.

Clarity:
The notation is described well, but it's not terribly intuitive (the query embedding is denoted by c, the history embedding by a, etc.), making section 4.4. hard to follow. A figure would have made things easier to follow, esp. due to the complexity of the model. A clearer parallel with previous methods would also improve the paper: is the proposed approach adding GCN on top of an established pipeline? Why not?

More discussion on code-mixed language, e.g. in section 4.6, would also improve clarity a bit (make the paper more self-contained). While the concept is clear from the context, it would be helpful to describe the level of structure in the mixed language. For instance, can dependency trees not be obtained code-mixed languages? Is there any research in this direction? (or is the concept very new?) Maybe I am just missing the background here, but it seems helpful in order to asses how appropriate the selected heuristic (based on the co-occurence matrix) is.

Relevant Reference:
Learning Graphical State Transitions, Johnson, ICLR 2017 also uses graph representations in question answering, though in a somewhat different setting.

Typos:
Section 4: "a model with following components"
Section 5: "the various hyperparameters that we conisdered"

---

> ### Author Response · Authors · 2018-11-24
> **Authors’ response to Reviewer 1**
>
> We would like to thank you for some great suggestions on strengthening the paper. We must confess that while we had some of these on our to-do list, there were a few that we hadn't actually thought of. We have now been able to add these experiments and we believe it has definitely helped us improve the quality of the paper. Below we give a pointwise update about the new experiments.
>
> 1)Passing information across the KB tree and query/history tree by aligning query/history elements with the KB elements: We were able to implement this and did a thorough hyperparameter tuning across all languages. We have included these results in the paper (RNN+CROSS-GCN-SeA in Tables 1, 2) but the short summary is that there was not much change in the BLEU, ROUGE and per response accuracy and only a marginal improvement in the Entity F1-score for En-DSTC2 and Ta-DSTC2. We had expected the entity F1-score to improve significantly across all languages since we are explicitly linking entities in the KB with entities in the query/history but unfortunately this was not the case. Initial analysis suggests that given that the task is relatively simple, even the base model, which does not explicitly pass information across the trees, is still able to capture the relevant information.
>
> 2)Ablation tests including comparisons with basic RNN based models and basic attention models: This was a bad miss on our part but now we have been able to do a thorough ablation study with the following experiments where we try to evaluate the (i) need for GCNs (ii) need for our sequential attention mechanism and (iii) need for combining RNNs with GCN:
>
> a)RNN with attention (the basic seq2seq+attention model of Bahdanau et. al. 2015)
> b)GCN with Bahdanau attention [does not use RNN or our sequential attention]
> c)RNN+GCN with Bahdanau attention [does not use our sequential attention]
> d)RNN with our sequential attention [does not use GCNs]
> e)RNN+GCN with our sequential attention [Our Final Model]
>
> We have included these results for all languages in the updated version of the paper (see Table 8 in Appendix D and “Ablations” part of Section 6) and the main observations are summarized below:
>
> i)GCNs do not outperform RNNs independently: In general, the performance of GCN-Bahdanau attention < RNN-Bahdanau attention
> ii)Our sequential attention outperforms Bahdanau attention:  In general, the performance of GCN-Bahdanau attention < GCN-our_seq_attention, RNN-Bahdanau attention < RNN-our_seq_attention and RNN+GCN-Bahdanau attention < RNN+GCN-our_seq_attention. However, note that RNN-Bahdanau attention < RNN-our_seq_attention holds for BLEU and all ROUGE metrics but not for Entity F1 and exact match accuracy. We are analyzing this further and will hopefully be able to add some insights in the final version of the paper.
> iii)Combining GCNs with RNNs helps: In general, RNN-our_seq_attention < RNN+GCN-our_seq_attention
>
> Overall, the best results are always obtained by our final model which combines RNN, GCN and sequential attention. Also, the code for our model and these ablation studies will be made publicly available.
>
> 3)Motivation behind attention: The motivation behind using a sequential attention mechanism was as follows: The current utterance which we refer to as query sets the stage for what comes next (the response). Hence we use this query to attend to only important parts in the history (essentially, the history can be long and we just want to focus on things which are relevant for the last utterance). Once, we have identified relevant portions of the history and computed an attention weighted representation for the history we are now ready to identify the important concepts from the KB. To achieve this effect we use the sequential attention mechanism.
>
> 4)GCN on top of an established pipeline: experiment c in point 2 above.
>
> 5)Better notations and figures: Indeed, in hindsight, we agree that some of our choices were not very intuitive. We have added 2 diagrams which hopefully makes things clear. It would be great if you can give us a feedback on the diagrams.
>
> 6)Clarity on code-mixing: The statistics about the level of code mixing, level of structure, etc are mentioned in the original paper (Banerjee et. al. 2018) which introduced the dataset. As suggested, to make the paper self-contained we have added the important statistics in this paper and some examples of code mixed conversations from the dataset (Appendix A). Note that there is a lot of work on processing code mixed text (for example, POS tagging of code mixed text, sentiment analysis of code mixed text, information retrieval using code mixed queries, etc). However, there is not much work on code mixed dialogues because this dataset was only released recently (COLING 2018). To the best of our knowledge, there is no work on building parsers for code mixed languages which produce parse trees.
>
> 7)We have fixed the typos and added the relevant reference.

---

### Meta-Review · Area_Chair1 · 2018-11-07
**Good performance but not much novelty**

**Confidence:** 4
**Recommendation:** Reject

**Metareview:**

This paper describes a graph convolutional network (GCN) approach to capture relational information in natural language as well as knowledge sources for goal-oriented dialogue systems. Relational information is captured by dependency parses, and when there is code switching in the input language, word co-occurrence information is used instead. Experiments on the modified DSTC2 dataset show significant improvements over baselines.
The original version of the paper lacked comparison to some SOTA baselines as also raised by the reviewers, these are included in the revised version.
Although the results show improvements over other approaches, it is arguable BLEU and ROUGE scores are not good enough for this task. Inclusion of human evaluation in the results would be very useful.